# RobustSpring: Benchmarking Robustness to Image Corruptions for Optical Flow, Scene Flow and Stereo

Jenny Schmalfuss*    Victor Oei*    Lukas Mehl    Madlen Bartsch

Shashank Agnihotri    Margret Keuper    Andres Bruhn

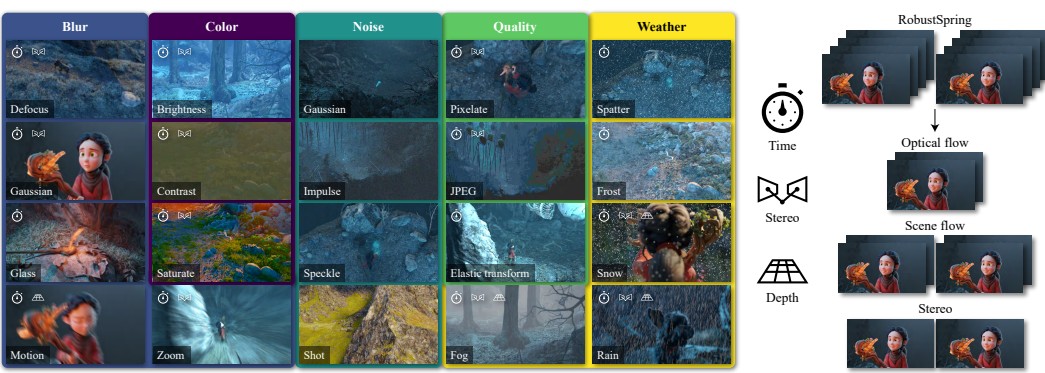

Figure 1: RobustSpring is a novel image corruption benchmark for optical flow, scene flow and stereo. It evaluates 20 image corruptions including blurs, color changes, noises, quality degradations, and weather, applied to stereo video data from [39]. For comprehensive robustness evaluations on all three tasks, RobustSpring's image corruptions are integrated in time, stereo and depth where applicable.

## Abstract

Standard benchmarks for optical flow, scene flow, and stereo vision algorithms generally focus on model accuracy rather than robustness to image corruptions like noise or rain. Hence, the resilience of models to such real-world perturbations is largely unquantified. To address this, we present RobustSpring, a comprehensive dataset and benchmark for evaluating robustness to image corruptions for optical flow, scene flow, and stereo models. RobustSpring applies 20 different image corruptions, including noise, blur, color changes, quality degradations, and weather distortions, in a time-, stereo-, and depth-consistent manner to the high-resolution Spring dataset, creating a suite of 20,000 corrupted images that reflect challenging conditions. RobustSpring enables comparisons of model robustness via a new corruption robustness metric. Integration with the Spring benchmark enables public two-axis evaluations of both accuracy and robustness. We benchmark a curated selection of initial models, observing that robustness varies widely by corruption type and experimentally show that evaluations on RobustSpring indicate real-world robustness. RobustSpring is a new computer vision benchmark at https://spring-benchmark.org that treats robustness as a first-class citizen to foster models that combine accuracy with resilience.

---

*Equal Contribution

Submitted to 39th Conference on Neural Information Processing Systems (NeurIPS 2025). Do not distribute.

# 1 Introduction

Optical flow, scene flow, and stereo vision algorithms estimate dense correspondences and enable real-world applications like robot navigation [36, 76, 30], video processing [40], structure-from-motion [34, 46], medical image registration [43] or surgical assistance [52, 47]. While estimation quality continuously improves on accuracy-driven benchmarks [39, 41, 8, 6, 53, 14, 51, 58], their robustness to real-world visual corruptions like sensor noise or compression artifacts is rarely systematically assessed. This lack of systematic assessment is problematic, as better accuracy does not necessarily translate to improved robustness and can even harm model robustness [67, 56]. Though image data in KITTI [41], Sintel [8] or Spring [39] comes with degradations like motion blurs, depth-of-field or brightness changes, they result from real-world data capture or efforts to increase data realism, but were not included to systematically study model predictions under image corruptions. Broad corruption-robustness studies as they exist for for image classification [17, 44], 3D object detection [42, 28] or monocular depth estimation [26] are rare for dense-correspondence tasks, where studies are limited to specific degradations like weather [57] or low-light [78]. This not only leaves uncertainty about the reliability of dense matching algorithms in real-world scenarios. It also prevents systematic efforts to improve their robustness.

To enable systematic studies on the image corruption robustness of optical flow, scene flow, and stereo, we propose the *RobustSpring* dataset. Based on Spring [39], it jointly benchmarks robustness of all three tasks on corrupted stereo videos. While prior image corruptions affect the monocular 2D or 3D space [17, 26, 42], RobustSpring's image corruptions are integrated in *time*, *stereo* and *depth* and thus tailored to dense matching tasks. A principled corruption robustness metric and public benchmark website make RobustSpring the first systematic tool to evaluate and improve dense matching robustness to image corruptions.

**Contributions.** Figure 1 gives an overview of RobustSpring. In summary, we make the following contributions:

(1) *Tailored image corruptions.* RobustSpring is the first image corruption dataset for optical flow, scene flow and stereo. It integrates 20 corruptions for blurs, noises, tints, artifacts, and weather in time, stereo, and depth.

(2) *Corruption robustness metric.* We propose a corruption robustness metric, based on Lipschitz continuity, which subsamples the clean-corrupted prediction difference and disentangles robustness and accuracy.

(3) *Benchmark functionality.* RobustSpring's standardized evaluation enables community-driven robustness comparisons of dense matching models. Public robustness benchmarking can be integrated with Spring's website.

(4) *Initial robustness evaluation.* We benchmark eight optical flow, two scene flow and six stereo models. All models are corruption sensitive, which reveals concealed robustness deficits on dense matching models.

**Intended Use.** RobustSpring is not a fine-tuning dataset, but a benchmark of how dense matching models generalize to *unseen* image corruptions. It seeks to foster robustness research and, simultaneously, helps assess real-world applicability of models. Hence, it is essential to tie RobustSpring to an existing accuracy benchmark like Spring, as this minimizes the robustness evaluation hurdle for researchers.

# 2 Related Work

While the quality of optical flow, scene flow and stereo models advanced for over three decades, their robustness recently regained attention as result of brittle deep learning generalization [49, 56]. We review robustness in dense-matching, particularly image corruptions and metrics.

**Robustness in Dense Matching.** Robustness research for optical flow, scene flow, and stereo models often focuses on *adversarial attacks*, which quantify prediction errors for optimized image perturbations. Most attacks are for optical flow [4, 57, 56, 59, 49, 73, 29] rather than stereo [7, 70] and scene flow [68, 33]. As remedies to adversarial vulnerability [3, 2, 1, 59, 5] may be overcome through specialized optimization [54], another line of robustness research considers unoptimized

data shifts. Those come in two flavors: *generalization across datasets*, *i.e.* the Robust Vision Challenge [http://www.robustvision.net/], and *robustness to image corruptions*. Dense matching models typically report generalization [38, 65, 66, 32, 19, 72] to several datasets, which span synthetic [39, 8, 51, 35, 11, 13, 50, 31] and real-world data [14, 41, 27, 53, 58], often in automotive contexts. While some datasets contain image corruptions, *e.g.* motion blur, depth of field, fog, noise or brightness changes [62, 8, 39, 41], they do not systematically assess corruption robustness. Yet, in the wild, robustness to image corruptions is crucial. For optical flow, systematic low light [78] and weather datasets [55, 57] exist, and [59, 74] apply 2D image corruptions [17] to optical flow data. Beyond these isolated works on optical flow, no systematic image-corruption study before RobustSpring spans all three dense matching tasks and includes scene flow or stereo.

**Robustness to Image Corruptions.** Popularized by 2D common corruptions [17], the field of image corruption robustness rapidly expanded from classification [17, 44] to depth estimation [26], 3D object detection [42, 28] and semantic segmentation [28]. Conceptually, corruptions were extended to the 3D space [26], LiDAR [28] and procedural rendering [12], but none have been tailored to the depth-, stereo-, and time-dependent setup of dense matching with optical flow, scene flow and stereo.

**Robustness Metrics and Benchmarks.** Most robustness metrics for dense matching differ by whether they utilize ground truth [49, 4, 74] or not [56, 57, 55]. However, multiple works [56, 67, 64] evidence that robustness and accuracy are competing qualities whose quantification should not be mixed, which informs our robustness metric. RobustSpring is the first dense-matching *robustness* benchmark, and joins prior classification robustness benchmarks [10, 25, 63]

# 3 RobustSpring Dataset and Benchmark

RobustSpring is a large, novel, image corruption dataset for optical flow, scene flow, and stereo. Below, we describe how we build on Spring's stereo video dataset and augment its frames with diverse image corruptions integrated in time, stereo, and depth, how we evaluate robustness to image corruptions, and use it to benchmark algorithm capabilities.

**Spring Data.** Spring [39] is a high-resolution benchmark and dataset with rendered stereo sequences. It is the ideal base for an image corruption dataset as its detailed renderings permit image alterations of varying granularity – from removing detail by blurring to adding detail via weather. Being a benchmark, Spring has a public training and closed test split, which withholds ground truth for optical flow, disparity, and extrinsic camera parameters. Because our robustness benchmark shall complement accuracy analyses, we use the 2000 Spring test frames, two per stereo camera. For image corruptions with time, stereo, and depth consistency, however, we require the extrinsic camera parameters and depths that are withheld. Thus, we estimate extrinsics using COLMAP 3.8 and depths as $Z = \frac{f_x \cdot B}{d}$, with focal length $f_x$, baseline length $B$ and stereo disparities $d$, estimated via MS-RAFT+ [22, 23]. Estimation also prevents data leakage and maintains ground truth confidentiality.

## 3.1 Corruption Dataset Creation

RobustSpring corrupts the Spring test frames via 20 diverse image corruptions, summarized in Fig. 2a and Fig. 2b. Below, we describe the image corruption types, their new consistencies, their implementation, and their severity levels.

**Corruption Types.** In RobustSpring, we consider the five image corruption types from [17]: color, blur, noise, quality, and weather. Color simulates different lighting conditions and camera settings, including brightness, contrast, and saturation. Blur acts like focus and motion artifacts, including defocus, Gaussian, glass, motion, and zoom blur. Noise represents sensor errors and ambiance, including Gaussian, impulse, speckle, and shot noise. Quality distortions are lossy compressions and geometric distortions, including pixelation, JPEG, and elastic transformations. Weather enacts outdoor conditions, including spatter, frost, snow, rain, and fog. All corruptions are on a single frame in Fig. 2a.

**Corruption Consistencies.** To increase the realism of these 20 corruptions for dense matching models, we extend their definition to time, stereo, and depth: *Time consistent* corruptions are smooth

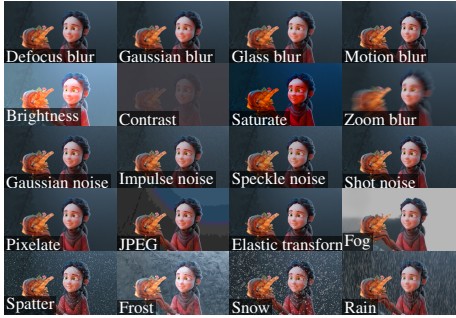

| | Color | | | Blur | | | | | Noise | | | | Qual | | | Weather | | | | |
|---|---|---|---|---|---|---|---|---|---|---|---|---|---|---|---|---|---|---|---|---|
| Property | Brightness | Contrast | Saturate | Defocus | Gaussian | Glass | Motion | Zoom | Gaussian | Impulse | Speckle | Shot | Pixelate | JPEG | Elastic | Spatter | Frost | Snow | Rain | Fog |
| Time-cons. | ✓ | ✓ | ✓ | ✓ | ✓ | ✓ | ✓ | ✓ | – | – | – | – | ✓ | ✓ | ✓ | ✓ | ✓ | ✓ | ✓ | ✓ |
| Stereo-cons. | ✓ | ✓ | ✓ | ✓ | ✓ | – | – | ✓ | – | – | – | – | ✓ | ✓ | – | – | – | ✓ | ✓ | ✓ |
| Depth-cons. | – | – | – | – | – | – | ✓ | – | – | – | – | – | – | – | – | – | – | ✓ | ✓ | ✓ |
| SSIM | 0.70 | 0.70 | 0.72 | 0.70 | 0.70 | 0.73 | 0.75 | 0.70 | 0.20 | 0.20 | 0.20 | 0.22 | 0.70 | 0.70 | 0.70 | 0.72 | 0.73 | 0.70 | 0.71 | |

(a) Image corruptions on a single image.

(b) Overview of corruptions and their consistency in time, stereo or depth, with resulting visual changes w.r.t. the original images as SSIM.

Figure 2: Overview of RobustSpring's image corruptions.

over time on *one* camera, *e.g.* frost on a camera lens, which differs per stereo camera. *Stereo consistent* corruptions equally influence both stereo cameras, *e.g.* brightness changes affect the cameras to the same extent. *Depth consistent* corruptions are integrated into the 3D scene, *e.g.* snowflakes falling along a trajectory in the 3D space, rendered into the camera view. Fig. 2b summarizes the consistencies we added to 16 of our 20 corruptions. Note that depth-aware motion blur is not stereo-consistent because it depends on the specific camera view.

**Corruption Implementation.** Though most corruptions are loosely based on [17], our corruption consistencies requires multiple adaptations. Furthermore, we employ specialized techniques for highly consistent effects, *i.e.* motion blur, elastic transform, snow, rain and fog. We adapt implementations from [17], modify glass blur, zoom blur, frost and pixelation to accommodate higher resolutions and non-square images, and adjust frost, glass blur, and spatter for consistency across video scenes. Motion blur is based on [77] and adds camera-induced motion with clean optical flow estimates. Elastic transform uses PyTorch's transforms package to create a see-through water-like effect, changing object morphology with smooth frame transitions. For snow and rain, we expand [57]'s two-step 3D particle rendering to multi-step particle trajectories and stereo views, change from additive-blending to order-independent alpha blending [37], and include global illumination [15]. To augment the large-scale Spring data, we improve its performance via more effective particle generation and parallel processing. Fog is based on the Koschmieder model following [69]. Full implementation details are in the supplementary.

**Corruption Severity.** Prior works [17, 44, 26, 42, 28] defined corruptions with several levels of severity. Here we opt for one severity per corruption, because evaluating one scene flow model on all 20 corruptions already produces 2.1 TB of raw data – 1.2 GB after subsampling, *c.f.* Sec. 3.2. More severity levels would overburden the evaluation resources of RobustSpring benchmark users. To balance severity across corruptions, we tune their hyperparameters until the image SSIM reaches a defined threshold. We generally use SSIM $\geq 0.7$, and, because the SSIM is less sensitive to blurs than noises [18], SSIM $\geq 0.2$ for noises for visually similar artifact strengths. Final SSIMs are in Fig. 2b.

### 3.2 Robustness Evaluation Metric

With various corruption types, we need a metric to quantify model robustness to these variations. In the following, we motivate and derive a ground-truth-free robustness metric for dense matching, introduce subsampling for efficiency, and discuss strategies for joint rankings over corruptions.

**Robustness Metric Concepts.** For dense matching, robustness to corruptions is undefined. Metrics exist for adversarial robustness, using the distance between corrupt prediction and either (i) ground-truth [49, 4] or (ii) clean prediction [56, 57, 55]. The latter is preferred for two reasons: First, (i)'s ground-truth comparisons mix accuracy and robustness, which are competing model qualities [56, 67, 64] that should be separate. This competition is intuitive: A model that always outputs the same value is as robust as inaccurate. Likewise, an accurate model varies for any input change and thus is not robust. Second, (ii) separates robustness from accuracy and builds on an established mathematical concept for system robustness [16, 45]: the Lipschitz constant $L^c$. It defines robust models as those

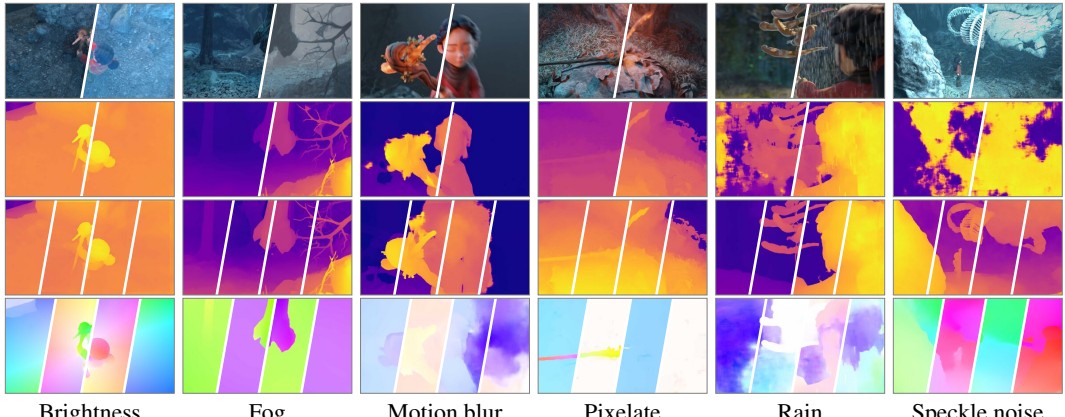

| Brightness | Fog | Motion blur | Pixelate | Rain | Speckle noise |

Figure 3: RobustSpring example frames. The first row shows clean and corrupted images. The second row shows the left and right disparity maps predicted with LEA Stereo [9]. The third row shows the target disparities for forward left, backward left, forward right, and backward right directions from M-FUSE [38]. The fourth row shows optical flow estimates for forward left, backward left, forward right, and backward right from RAFT [65]. All disparities and flows are computed on the corrupted dataset, see supplementary for additional frames.

whose prediction $f$ is similar on clean and corrupt image $I$ and $I^c$, relative to their difference. For dense matching, it reads

$$L^c = \frac{\|f(I) - f(I^c)\|}{\|I - I^c\|}. \tag{1}$$

This robustness formulation is preferable for real-world applications that demand stable scene estimations *despite* corruptions like snow.

**Corruption Robustness Metric.** Based on Eq. (1), we quantify model robustness to corruptions. Because RobustSpring's corrupt images $I_c$ deviate from their clean counterparts $I$ by a similar amount, *c.f.* SSIM equalization in Sec. 3.1, we omit the denominator in Eq. (1) and define *corruption robustness* as distance between clean $f(I)$ and corrupted $f(I^c)$ predictions with distance metric M:

$$R_M^c = M[f(I), f(I^c)]. \tag{2}$$

For similarity to Spring's evaluation, we use corruption robustness with various metrics M, reporting $R_{EPE}^c$, $R_{1px}^c$ and $R_{Fl}^c$ for optical and scene flow, and $R_{1px}^c$, $R_{Abs}^c$ and $R_{D1}^c$ for stereo. Interestingly, our EPE-based corruption robustness

$$R_{EPE}^c = EPE[f(I), f(I^c)] = \frac{1}{|\Omega|} \sum_{i \in \Omega} \|f_i(I) - f_i(I^c)\|, \tag{3}$$

on image domain $\Omega$ is a generalization of optical-flow adversarial robustness [56] to dense matching and corruptions.

**Metric Subsampling.** For a benchmark, users should upload robustness results to a web server. Given the large number of 20 datasets, data reduction is essential to facilitate evaluations and uploads. To this end, we evaluate on a reduced set of pixels by refining the original subsampling strategy from Spring, which retains about 1% of the full data. First, we additionally subsample the set of full-resolution Hero-frames, leaving 0.95%, and then apply 20-fold subsampling, ultimately keeping 0.05% of the full data.

**Robustness Ranking.** Because we generate 20 different corruption evaluations *per* dense matching model, we need a summarization strategy to produce one result per model. Per-model results are ranked based on three strategies: Average, Median, and the Schulze voting method [60]. In contrast to averaging across all 20 evaluations, the median reduces the impact of extreme outliers. The Schulze method provides a holistic, pairwise comparison approach that ranks models based on preference aggregation and was used for prior generalization evaluations in the Robust Vision Challenges. We evaluate their differences in Sec. 4.2

Table 1: Initial RobustSpring results on corruption robustness of optical flow models, using $R^c_{\text{EPE}}$, $R^c_{\text{1px}}$ and $R^c_{\text{Fl}}$ between clean and corrupted flow predictions. Low values indicate robust models. *Clean Error* compares clean predictions and ground-truth flows, values from [39].

| | | GMFlow | | | MS-RAFT+ | | | FlowFormer | | | GMA | | | SPyNet | | | RAFT | | | FlowNet2 | | | PWCNet | | |
|---|---|---|---|---|---|---|---|---|---|---|---|---|---|---|---|---|---|---|---|---|---|---|---|---|---|
| | | $R^c_{\text{EPE}}$ | $R^c_{\text{1px}}$ | $R^c_{\text{Fl}}$ | $R^c_{\text{EPE}}$ | $R^c_{\text{1px}}$ | $R^c_{\text{Fl}}$ | $R^c_{\text{EPE}}$ | $R^c_{\text{1px}}$ | $R^c_{\text{Fl}}$ | $R^c_{\text{EPE}}$ | $R^c_{\text{1px}}$ | $R^c_{\text{Fl}}$ | $R^c_{\text{EPE}}$ | $R^c_{\text{1px}}$ | $R^c_{\text{Fl}}$ | $R^c_{\text{EPE}}$ | $R^c_{\text{1px}}$ | $R^c_{\text{Fl}}$ | $R^c_{\text{EPE}}$ | $R^c_{\text{1px}}$ | $R^c_{\text{Fl}}$ | $R^c_{\text{EPE}}$ | $R^c_{\text{1px}}$ | $R^c_{\text{Fl}}$ |
| **Color** | Brightness | 0.33 | 3.31 | 1.12 | 0.33 | 2.88 | 1.02 | 0.68 | 2.82 | 1.05 | 0.36 | 3.22 | 1.04 | 2.72 | 14.67 | 8.91 | 0.92 | 3.49 | 1.61 | 0.45 | 3.16 | 1.05 | 1.04 | 7.38 | 3.00 |
| | Contrast | 0.46 | 6.71 | 1.71 | 0.87 | 6.69 | 3.24 | 0.93 | 5.48 | 1.96 | 0.68 | 6.43 | 2.20 | 8.23 | 38.90 | 27.23 | 1.32 | 5.73 | 2.64 | 1.87 | 9.26 | 4.74 | 2.98 | 30.07 | 7.42 |
| | Saturate | 0.34 | 3.30 | 0.96 | 0.34 | 2.87 | 1.03 | 0.42 | 2.39 | 0.88 | 0.43 | 3.47 | 1.18 | 3.36 | 17.34 | 11.31 | 0.93 | 3.33 | 1.47 | 0.51 | 3.40 | 1.10 | 1.21 | 9.92 | 3.68 |
| **Blur** | Defocus | 0.53 | 6.17 | 1.45 | 0.51 | 4.01 | 1.47 | 0.55 | 3.85 | 1.19 | 0.56 | 5.02 | 2.01 | 0.57 | 10.16 | 1.36 | 1.03 | 4.70 | 2.07 | 0.53 | 3.35 | 1.06 | 0.98 | 6.51 | 2.78 |
| | Gaussian | 0.66 | 7.77 | 1.88 | 0.58 | 4.45 | 1.63 | 0.63 | 4.32 | 1.37 | 0.62 | 5.48 | 2.22 | 0.76 | 15.44 | 2.12 | 1.10 | 5.12 | 2.26 | 0.60 | 4.05 | 1.27 | 1.11 | 7.72 | 3.09 |
| | Glass | 0.85 | 20.87 | 1.82 | 0.53 | 4.45 | 1.37 | 0.64 | 4.04 | 1.17 | 0.61 | 5.60 | 1.91 | 0.75 | 16.94 | 1.36 | 1.05 | 5.13 | 1.97 | 0.50 | 3.12 | 0.96 | 0.91 | 5.96 | 2.47 |
| | Motion | 1.34 | 18.35 | 7.51 | 1.31 | 14.06 | 6.16 | 1.35 | 14.03 | 5.77 | 1.19 | 14.40 | 6.18 | 2.32 | 19.55 | 10.05 | 2.06 | 14.33 | 6.35 | 1.60 | 14.07 | 6.47 | 1.95 | 16.25 | 7.47 |
| | Zoom | 1.88 | 35.80 | 9.90 | 1.81 | 21.84 | 7.13 | 1.66 | 22.72 | 6.77 | 1.54 | 23.17 | 7.16 | 4.82 | 46.67 | 28.37 | 3.14 | 22.80 | 7.61 | 2.36 | 24.63 | 9.04 | 3.52 | 50.33 | 15.64 |
| **Noise** | Gaussian | 4.70 | 57.95 | 21.67 | 5.70 | 35.74 | 22.12 | 6.56 | 27.83 | 18.30 | 2.81 | 24.70 | 12.96 | 2.22 | 42.23 | 14.88 | 7.43 | 27.92 | 18.99 | 1.33 | 11.24 | 5.06 | 2.79 | 26.87 | 9.89 |
| | Impulse | 6.64 | 66.14 | 28.70 | 7.39 | 45.72 | 29.05 | 7.33 | 23.58 | 14.47 | 4.08 | 31.31 | 18.13 | 2.92 | 53.45 | 20.41 | 6.51 | 29.65 | 18.32 | 2.37 | 15.70 | 7.48 | 3.57 | 35.67 | 14.45 |
| | Speckle | 3.90 | 62.01 | 20.64 | 4.22 | 34.96 | 17.18 | 5.47 | 25.52 | 15.60 | 5.32 | 25.22 | 12.66 | 1.95 | 46.32 | 12.89 | 6.62 | 26.05 | 16.48 | 1.32 | 12.57 | 4.19 | 2.74 | 26.83 | 8.00 |
| | Shot | 3.52 | 56.71 | 17.77 | 4.36 | 31.67 | 17.77 | 5.75 | 26.02 | 16.01 | 3.15 | 23.11 | 11.59 | 1.86 | 40.44 | 11.98 | 6.74 | 25.64 | 17.08 | 1.16 | 9.87 | 3.92 | 2.59 | 23.75 | 7.88 |
| **Quality** | Pixelate | 1.96 | 68.09 | 18.71 | 1.60 | 45.83 | 6.78 | 1.48 | 31.68 | 2.59 | 1.11 | 25.86 | 1.78 | 1.22 | 50.63 | 2.90 | 1.65 | 21.47 | 2.00 | 0.77 | 7.74 | 0.88 | 0.92 | 8.67 | 2.22 |
| | JPEG | 3.32 | 83.54 | 27.92 | 2.09 | 41.69 | 12.82 | 2.89 | 42.62 | 14.96 | 1.92 | 38.70 | 11.51 | 2.95 | 53.97 | 18.08 | 3.19 | 37.72 | 13.67 | 2.56 | 31.00 | 11.85 | 2.88 | 49.15 | 15.91 |
| | Elastic | 1.37 | 40.00 | 6.89 | 1.16 | 32.49 | 5.54 | 2.62 | 35.78 | 11.01 | 1.24 | 27.24 | 6.40 | 1.08 | 34.62 | 4.77 | 1.33 | 19.43 | 4.78 | 0.79 | 16.27 | 2.12 | 1.42 | 28.18 | 5.47 |
| **Weather** | Fog | 0.80 | 14.42 | 5.32 | 0.91 | 10.32 | 6.33 | 0.86 | 9.66 | 5.67 | 0.84 | 11.21 | 6.42 | 5.20 | 28.15 | 19.97 | 1.97 | 12.01 | 7.11 | 1.74 | 11.77 | 7.82 | 16.84 | 20.96 | 12.89 |
| | Frost | 8.20 | 63.96 | 29.96 | 7.38 | 29.96 | 21.25 | 8.18 | 34.19 | 23.87 | 8.13 | 34.30 | 22.31 | 6.97 | 45.13 | 30.13 | 8.37 | 32.75 | 21.76 | 7.22 | 33.69 | 21.15 | 8.27 | 50.31 | 27.44 |
| | Rain | 8.60 | 64.20 | 32.72 | 19.99 | 36.74 | 31.22 | 11.13 | 33.50 | 20.83 | 33.00 | 43.98 | 36.18 | 18.20 | 68.87 | 56.38 | 42.41 | 38.89 | 31.99 | 63.71 | 48.25 | 41.15 | 40.18 | 73.51 | 57.05 |
| | Snow | 3.60 | 70.60 | 29.90 | 4.69 | 33.21 | 30.91 | 7.92 | 40.20 | 33.82 | 5.30 | 40.82 | 33.35 | 12.08 | 74.27 | 66.65 | 7.16 | 37.04 | 31.37 | 39.79 | 68.67 | 61.60 | 39.73 | 90.80 | 81.91 |
| | Spatter | 6.58 | 67.90 | 27.09 | 6.63 | 28.22 | 20.24 | 8.41 | 40.38 | 26.92 | 7.75 | 36.11 | 21.81 | 5.71 | 48.60 | 33.82 | 7.98 | 30.37 | 19.87 | 9.13 | 45.03 | 28.99 | 9.33 | 65.41 | 40.19 |
| **Average** | | 2.98 | 40.89 | 14.68 | 3.62 | 23.39 | 12.21 | 3.77 | 21.53 | 11.21 | 4.03 | 21.47 | 10.95 | 4.29 | 38.32 | 19.18 | 5.64 | 20.18 | 11.47 | 7.01 | 18.84 | 11.09 | 7.25 | 31.71 | 16.44 |
| Std. Dev. | | 2.70 | 27.91 | 11.91 | 4.58 | 15.54 | 10.62 | 3.44 | 14.37 | 9.94 | 7.23 | 13.67 | 10.55 | 4.38 | 18.35 | 17.60 | 9.10 | 12.55 | 9.98 | 15.94 | 17.93 | 15.87 | 11.83 | 24.43 | 20.79 |
| **Median** | | 1.92 | 48.35 | 13.83 | 1.71 | 29.09 | 6.95 | 2.14 | 24.55 | 8.89 | 1.39 | 23.93 | 6.79 | 2.82 | 41.33 | 13.88 | 2.60 | 22.13 | 7.36 | 1.47 | 12.17 | 4.90 | 2.77 | 26.85 | 7.94 |
| Clean Error | | 0.94 | 10.36 | 2.95 | 0.64 | 5.72 | 2.19 | 0.72 | 6.51 | 2.38 | 0.91 | 7.07 | 3.08 | 4.16 | 29.96 | 12.87 | 1.48 | 6.79 | 3.20 | 1.04 | 6.71 | 2.82 | 2.29 | 82.27 | 4.89 |

## 3.3 Dataset and Benchmark Functionality

Below, we summarize RobustSpring's corruption dataset and describe its benchmark function. Fig. 3 shows data samples with stereo, optical flow and scene flow estimates.

**RobustSpring Dataset.** The final RobustSpring dataset entails 20 corrupted versions of Spring, resulting in 40,000 frames, or 20,000 stereo frame pairs. Each corruption evaluation yields 3960 optical flows (990 per camera & direction), 2000 stereo disparities (1000 per camera) and 3960 additional scene flow disparity maps (990 per camera per direction). We publicly release the RobustSpring test set licensed with CC BY 4.0, but no corrupt training data to discourage corruption finetuning for a fair benchmark. We separately provide the raw data and a curated dataset for predicting dense matches.

**RobustSpring Benchmark.** RobustSpring enables uploading robustness results to a benchmark website for display in a public ranking. To emphasize that robustness and accuracy are two axes of model performance with equal importance [67], we couple RobustSpring with Spring's established accuracy benchmark. Thus, researchers can report model robustness and accuracy on the same dataset. To maintain Spring's upload policy, 3 per 30 days, one per hour, each submission receives one robustness upload.

## 4 Results

We evaluate RobustSpring under two aspects: First, we report initial results for 16 optical flow, scene flow and stereo models. Then, we analyze the benchmark evaluation, particularly subsampling strategy and ranking methods.

### 4.1 Initial RobustSpring Benchmark Results

We provide initial results on RobustSpring for selected models from all three dense matching tasks. For optical flow, we include GMFlow [72], MS-RAFT+ [23], FlowFormer [19], GMA [24], SPyNet [48], RAFT [65], FlowNet2 [20], and PWCNet [61]. For scene flow, we evaluate M-FUSE [38] and RAFT-3D [66]. For stereo estimation, we evaluate RAFT-Stereo [32], ACVNet [71], LEAStereo [9], and GANet [75]. An overview of all models and used checkpoints is in the supplement. Importantly, none of these models are fine-tuned to either Spring or RobustSpring data, to assess the generalization capacity of existing models.

**Optical Flow.** The evaluation results in Tab. 1 show considerable robustness variations over the different corruption types, which we also visualize in Fig. 4a. Weather-based corruptions, especially

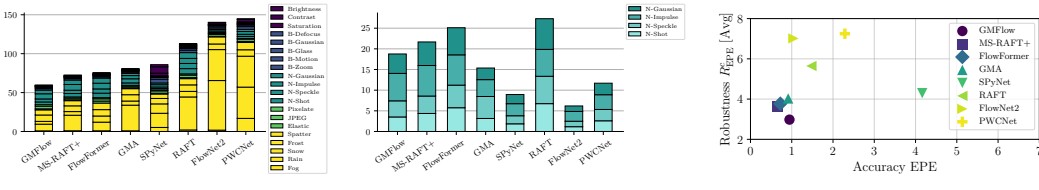

(a) All Corruptions  (b) Noise-Based Corruptions.  (c) Accuracy vs. Robustnes.

Figure 4: Accumulated corruption robustness $R_{\text{EPE}}^c$ for optical flow models over all corruptions *[left]*, only noise corruptions *[middle]*, and accuracy vs. robustness *[right]*. All other corruption classes color (purple), blur (blue), noise (cyan), quality (green), and weather (yellow) are in the supplement. Small values are robust (and accurate) models. The supplement shows accuracy vs. Median $R_{\text{EPE}}^c$.

rain and snow, degrade the performance most and lead to the largest $R^c$ values. In contrast, color-based corruptions have a relatively small impact, as most models maintain low $R_{\text{EPE}}^c$ values. Also, the order of models can change significantly depending on the corruption type. While FlowNet2 does not perform well in the overall comparison, it is the best model for noise-based corruptions in Figure 4b. Overall, GMFlow achieves the lowest average $R_{\text{EPE}}^c$, GMA the lowest median. We will detail on ranking differences in Sec. 4.2.

To investigate a potential accuracy-robustness tradeoff on image corruptions, we visualize both quantities in Fig. 4c. Overall, accurate models tend to be more robust, though we find a slight tradeoff because there is no unanimous winner in both dimensions – similarly for median robustness in the supplement. Interestingly, this contrasts with adversarial robustness evaluations, which observed a clear accuracy-robustness tradeoff on optical flow [56]. Potentially, this tradeoff is less pronounced for image corruptions as they are not optimized per model like adversarial attacks.

Focusing on the architecture of optical flow models, we find that transformer-based models, such as GMFlow and FlowFormer, generally outperform other architectures. However, they tend to struggle with noise corruptions, potentially resulting from their global matching. Hierarchical models, such as MS-RAFT+, achieve balanced performance for most corruptions and may benefit from multi-scale feature processing to cope with quality degradations. In contrast, stacked architectures such as FlowNet2 are uniquely resilient to noise, potentially due to their progressive refinement across layers. Overall, certain architectural features appear to influence robustness to certain corruption types, but there is no clear winner in terms of architecture.

**Scene Flow.** The results for scene flow are in Tab. 2a, and include optical flow and target frame disparity predictions for M-FUSE and RAFT-3D. M-FUSE generally produces more robust optical

Table 2: Initial RobustSpring results on corruption robustness of scene flow and stereo disparity models, using corruption robustness $R_{\text{1px}}^c$, $R_{\text{Abs}}^c$ and $R_{\text{Dl}}^c$ between clean and corrupted predictions. Low values indicate robust models. Corresponding Disparity 1 from scene flow models LEAStereo (s) for M-FUSE, and GANet (s) for RAFT-3D in Tab. 2b. Stereo disparity models use Stereo (s) and KITTI (k) checkpoints, *c.f.* supplementary.

(a) Initial scene flow evaluation.

| | | M-FUSE | | | | | | RAFT-3D | | | | | |
| | | Optical flow | | | Disparity 2 | | | Optical flow | | | Disparity 2 | | |
| | | $R_{\text{EPE}}^c$ | $R_{\text{1px}}^c$ | $R_{\text{F1}}^c$ | $R_{\text{Abs}}^c$ | $R_{\text{1px}}^c$ | $R_{\text{D2}}^c$ | $R_{\text{EPE}}^c$ | $R_{\text{1px}}^c$ | $R_{\text{F1}}^c$ | $R_{\text{Abs}}^c$ | $R_{\text{1px}}^c$ | $R_{\text{D2}}^c$ |
|---|---|---|---|---|---|---|---|---|---|---|---|---|---|
| Color | Brightness | 0.83 | 5.54 | 2.80 | 0.14 | 1.53 | 0.18 | 1.38 | 8.23 | 3.87 | 0.07 | 1.48 | 0.21 |
| | Contrast | 0.99 | 7.86 | 3.60 | 0.17 | 1.71 | 0.17 | 1.42 | 10.71 | 5.07 | 0.07 | 1.65 | 0.22 |
| | Saturate | 0.67 | 4.94 | 2.43 | 0.12 | 1.22 | 0.14 | 0.93 | 6.72 | 3.31 | 0.06 | 1.33 | 0.18 |
| Blur | Defocus | 0.84 | 5.26 | 2.71 | 0.15 | 1.37 | 0.15 | 0.66 | 5.27 | 2.44 | 0.04 | 0.88 | 0.10 |
| | Gaussian | 0.94 | 5.81 | 2.92 | 0.16 | 1.56 | 0.18 | 0.78 | 5.85 | 2.73 | 0.05 | 1.04 | 0.14 |
| | Glass | 0.80 | 5.17 | 2.65 | 0.16 | 1.32 | 0.14 | 0.65 | 5.29 | 2.39 | 0.04 | 0.82 | 0.09 |
| | Motion | 1.51 | 15.10 | 6.81 | 0.18 | 2.50 | 0.35 | 1.62 | 14.66 | 6.85 | 0.08 | 1.60 | 0.28 |
| | Zoom | 2.28 | 27.88 | 9.52 | 0.28 | 3.74 | 0.41 | 2.68 | 34.06 | 11.99 | 0.14 | 2.84 | 0.50 |
| Noise | Gaussian | 6.49 | 29.22 | 14.81 | 0.41 | 6.56 | 0.80 | 5.25 | 43.33 | 25.43 | 0.20 | 3.64 | 0.71 |
| | Impulse | 5.98 | 37.32 | 19.16 | 0.43 | 8.11 | 0.88 | 6.73 | 59.86 | 33.16 | 0.22 | 4.43 | 0.75 |
| | Speckle | 3.73 | 29.39 | 12.22 | 0.35 | 5.68 | 0.57 | 4.86 | 51.12 | 26.11 | 0.18 | 3.17 | 0.64 |
| | Shot | 4.87 | 26.32 | 12.34 | 0.36 | 5.60 | 0.69 | 4.65 | 42.07 | 22.91 | 0.18 | 3.26 | 0.67 |
| Quality | Pixelate | 0.86 | 5.95 | 2.51 | 0.19 | 1.51 | 0.13 | 0.82 | 7.66 | 2.83 | 0.05 | 1.02 | 0.10 |
| | JPEG | 1.98 | 27.21 | 6.82 | 0.32 | 3.62 | 0.36 | 2.73 | 33.93 | 10.55 | 0.13 | 2.59 | 0.41 |
| | Elastic | 1.15 | 14.93 | 3.92 | 0.22 | 2.28 | 0.22 | 1.70 | 21.82 | 5.99 | 0.08 | 1.61 | 0.20 |
| Weather | Fog | 2.35 | 15.39 | 10.13 | 0.19 | 2.43 | 0.19 | 2.29 | 18.15 | 11.67 | 0.06 | 1.23 | 0.15 |
| | Frost | 7.91 | 41.60 | 23.41 | 0.38 | 6.55 | 0.78 | 7.49 | 45.07 | 24.26 | 0.16 | 3.75 | 0.52 |
| | Rain | 10.21 | 41.78 | 28.99 | 0.70 | 12.79 | 1.29 | 27.89 | 74.23 | 59.77 | 0.47 | 10.75 | 1.96 |
| | Snow | 6.36 | 47.06 | 33.55 | 0.46 | 7.67 | 0.80 | 19.08 | 80.49 | 60.01 | 0.31 | 6.79 | 0.84 |
| | Spatter | 7.00 | 46.35 | 22.10 | 0.39 | 6.21 | 0.80 | 7.06 | 55.55 | 25.80 | 0.17 | 3.82 | 0.53 |
| **Average** | | 3.39 | 22.00 | 11.17 | 0.29 | 4.20 | 0.46 | 5.03 | 31.20 | 17.36 | 0.14 | 2.89 | 0.46 |
| Std. Dev. | | 2.95 | 15.23 | 9.60 | 0.15 | 3.11 | 0.34 | 6.85 | 24.26 | 17.63 | 0.11 | 2.40 | 0.43 |
| **Median** | | 2.13 | 20.86 | 8.17 | 0.25 | 3.06 | 0.35 | 2.49 | 27.88 | 11.11 | 0.10 | 2.12 | 0.35 |
| Clean Error | | 2.52 | 13.96 | 6.89 | 7.11 | 32.95 | 14.54 | 2.53 | 20.98 | 8.48 | 8.08 | 57.03 | 21.54 |

(b) Initial stereo disparity evaluation.

| | | RAFT-Stereo (s) | | | ACVNet (s) | | | LEAStereo (s) | | | LEAStereo (k) | | | GANet (k) | | | GANet (s) | | |
| | | $R_{\text{1px}}^c$ | $R_{\text{Abs}}^c$ | $R_{\text{Dl}}^c$ | $R_{\text{1px}}^c$ | $R_{\text{Abs}}^c$ | $R_{\text{Dl}}^c$ | $R_{\text{1px}}^c$ | $R_{\text{Abs}}^c$ | $R_{\text{Dl}}^c$ | $R_{\text{1px}}^c$ | $R_{\text{Abs}}^c$ | $R_{\text{Dl}}^c$ | $R_{\text{1px}}^c$ | $R_{\text{Abs}}^c$ | $R_{\text{Dl}}^c$ | $R_{\text{1px}}^c$ | $R_{\text{Abs}}^c$ | $R_{\text{Dl}}^c$ |
|---|---|---|---|---|---|---|---|---|---|---|---|---|---|---|---|---|---|---|---|
| Color | Brightness | 8.98 | 2.13 | 2.83 | 19.82 | 6.89 | 8.80 | 6.38 | 1.27 | 1.78 | 11.57 | 2.02 | 3.73 | 12.46 | 2.48 | 4.61 | 10.74 | 2.11 | 3.39 |
| | Contrast | 14.04 | 2.62 | 3.81 | 19.33 | 8.34 | 9.88 | 19.00 | 3.33 | 6.45 | 18.23 | 2.86 | 5.63 | 18.02 | 2.72 | 5.49 | 23.14 | 3.94 | 6.74 |
| | Saturate | 7.54 | 0.74 | 0.95 | 8.12 | 3.18 | 3.79 | 6.43 | 1.24 | 1.71 | 13.57 | 3.05 | 4.64 | 16.69 | 3.53 | 5.77 | 13.53 | 2.70 | 3.86 |
| Blur | Defocus | 10.61 | 2.47 | 3.90 | 8.06 | 1.10 | 1.90 | 8.55 | 2.02 | 2.49 | 29.31 | 3.26 | 5.21 | 41.32 | 3.29 | 4.68 | 12.34 | 2.46 | 3.16 |
| | Gaussian | 11.40 | 2.57 | 3.97 | 9.29 | 1.55 | 2.38 | 9.64 | 2.16 | 2.65 | 48.95 | 3.68 | 5.54 | 47.97 | 3.55 | 4.98 | 13.76 | 2.69 | 3.45 |
| | Glass | 13.10 | 2.61 | 3.34 | 11.72 | 1.31 | 1.95 | 11.56 | 2.17 | 2.55 | 70.01 | 4.79 | 6.36 | 71.45 | 4.33 | 5.18 | 19.42 | 2.61 | 3.15 |
| | Motion | 12.41 | 2.30 | 2.61 | 9.72 | 1.13 | 2.07 | 10.59 | 1.82 | 2.74 | 20.04 | 2.44 | 4.77 | 16.99 | 2.27 | 4.26 | 13.12 | 2.31 | 3.61 |
| | Zoom | 59.50 | 5.86 | 7.19 | 64.76 | 6.43 | 9.32 | 63.52 | 6.38 | 9.74 | 74.92 | 8.84 | 16.83 | 74.29 | 8.18 | 14.80 | 59.89 | 7.29 | 11.21 |
| Noise | Gaussian | 40.76 | 20.44 | 24.16 | 56.40 | 39.19 | 37.76 | 80.74 | 80.89 | 62.28 | 65.13 | 15.23 | 24.53 | 49.20 | 7.90 | 13.17 | 85.78 | 33.35 | 45.02 |
| | Impulse | 44.79 | 21.16 | 27.99 | 69.34 | 53.14 | 49.67 | 85.39 | 85.24 | 65.42 | 69.03 | 17.24 | 25.47 | 51.64 | 8.18 | 12.70 | 85.00 | 38.94 | 50.45 |
| | Speckle | 42.58 | 13.64 | 21.85 | 71.99 | 63.51 | 57.36 | 84.06 | 84.54 | 65.37 | 66.23 | 15.68 | 24.31 | 55.36 | 7.64 | 13.63 | 83.70 | 29.65 | 41.90 |
| | Shot | 39.84 | 15.55 | 20.23 | 59.56 | 42.20 | 41.10 | 79.41 | 76.53 | 59.94 | 64.06 | 14.29 | 22.95 | 49.36 | 6.95 | 11.98 | 81.49 | 28.20 | 39.89 |
| Quality | Pixelate | 66.69 | 46.19 | 13.86 | 57.29 | 4.14 | 4.98 | 35.19 | 3.85 | 4.11 | 57.19 | 3.72 | 4.83 | 62.71 | 4.00 | 4.60 | 59.61 | 3.70 | 4.07 |
| | JPEG | 55.27 | 8.24 | 5.27 | 60.87 | 15.98 | 15.16 | 55.18 | 9.20 | 10.84 | 68.22 | 5.63 | 7.97 | 65.92 | 7.41 | 11.19 | 59.52 | 6.76 | 10.10 |
| | Elastic | 65.53 | 6.52 | 4.32 | 58.39 | 8.17 | 7.29 | 71.96 | 8.02 | 10.92 | 93.40 | 7.16 | 8.90 | 87.38 | 6.89 | 8.86 | 76.47 | 4.85 | 5.05 |
| Weather | Fog | 13.71 | 1.57 | 2.10 | 17.99 | 17.70 | 12.12 | 17.95 | 14.25 | 10.88 | 23.36 | 8.18 | 12.90 | 21.36 | 9.69 | 12.45 | 20.55 | 9.68 | 9.75 |
| | Frost | 41.63 | 18.84 | 10.68 | 39.79 | 8.15 | 19.27 | 38.43 | 7.28 | 18.51 | 53.98 | 12.37 | 23.89 | 39.74 | 9.84 | 20.93 | 47.40 | 11.20 | 24.31 |
| | Rain | 43.10 | 79.42 | 32.27 | 34.62 | 12.92 | 18.48 | 56.55 | 22.14 | 34.58 | 65.45 | 12.54 | 28.62 | 49.08 | 11.44 | 22.55 | 59.22 | 26.50 | 42.34 |
| | Snow | 41.05 | 51.30 | 32.90 | 40.96 | 18.62 | 29.03 | 47.03 | 20.51 | 32.23 | 52.16 | 13.88 | 29.40 | 35.16 | 11.83 | 22.94 | 45.88 | 17.24 | 33.30 |
| | Spatter | 35.50 | 27.17 | 12.57 | 18.01 | 2.18 | 3.85 | 31.43 | 5.13 | 10.19 | 35.54 | 7.93 | 14.24 | 28.00 | 6.75 | 12.42 | 34.58 | 6.04 | 13.86 |
| **Average** | | 33.40 | 16.57 | 11.84 | 36.80 | 15.79 | 16.81 | 40.95 | 21.90 | 20.77 | 50.02 | 8.24 | 14.04 | 44.71 | 6.44 | 10.86 | 45.26 | 12.11 | 17.93 |
| Std. Dev. | | 20.16 | 20.72 | 10.79 | 23.56 | 18.64 | 17.08 | 29.19 | 31.33 | 23.64 | 23.69 | 5.15 | 9.43 | 21.37 | 3.00 | 6.10 | 28.09 | 12.17 | 17.25 |
| **Median** | | 40.30 | 7.38 | 6.23 | 37.21 | 8.16 | 9.60 | 36.81 | 6.83 | 10.51 | 55.58 | 7.55 | 10.90 | 48.53 | 6.92 | 11.58 | 46.64 | 6.40 | 9.93 |
| Clean | | 15.27 | 3.02 | 5.35 | 14.77 | 1.52 | 5.35 | 19.89 | 3.88 | 9.19 | 47.50 | 6.15 | 17.16 | 27.91 | 5.29 | 11.56 | 23.22 | 4.59 | 10.39 |

Table 3: Evaluations of the metrics used in RobustSpring.

(a) Influence of subsampling. We compare robustness evaluations on the full test data (Full) to evaluations on Spring's original subsampling (Spring), original subsampling without Hero-frames (Spring*), and our refined corruption subsampling (Ours).

(b) Robustness ranking of optical flow models with ranking strategies Average $R^c_{\text{EPE}}$, Median $R^c_{\text{EPE}}$, and Schulze to summarize results over corruptions. Please note that Schulze does not produce numeric values.

| | Subsampling $R^c_{\text{EPE}}$ | | | | Subsampling $R^c_{\text{1px}}$ | | | |
| --- | --- | --- | --- | --- | --- | --- | --- | --- |
| | Full | Spring | Spring* | Ours | Full | Spring | Spring* | Ours |
| % Original Data | 100% | 1.00% | 0.94% | 0.05% | 100% | 1.00% | 0.94% | 0.05% |
| GMFlow | 2.98 | 3.20 | 2.98 | 2.98 | 40.89 | 41.99 | 40.89 | 40.89 |
| MS-RAFT+ | 3.62 | 3.84 | 3.62 | 3.62 | 23.38 | 24.44 | 23.39 | 23.39 |
| FlowFormer | 3.77 | 3.89 | 3.77 | 3.77 | 21.52 | 22.39 | 21.53 | 21.53 |
| GMA | 4.03 | 4.28 | 4.03 | 4.03 | 21.47 | 22.59 | 21.48 | 21.47 |
| SPyNet | 4.30 | 4.56 | 4.29 | 4.29 | 38.32 | 39.28 | 38.32 | 38.32 |
| RAFT | 5.64 | 6.15 | 5.64 | 5.64 | 20.17 | 21.20 | 20.18 | 20.18 |
| FlowNet2 | 7.01 | 7.36 | 7.01 | 7.01 | 18.84 | 19.79 | 18.84 | 18.84 |
| PWCNet | 7.25 | 7.52 | 7.25 | 7.25 | 31.71 | 32.55 | 31.72 | 31.71 |

| | Ranking Method | | |
| --- | --- | --- | --- |
| Rank | Average $R^c_{\text{EPE}}$ | Median $R^c_{\text{EPE}}$ | Schulze |
| 1 | 2.98 GMFlow | 1.39 GMA | MS-RAFT+ |
| 2 | 3.62 MS-RAFT+ | 1.47 FlowNet2 | GMA |
| 3 | 3.77 FlowFormer | 1.71 MS-RAFT+ | FlowNet2 |
| 4 | 4.03 GMA | 1.92 GMFlow | GMFlow |
| 5 | 4.29 SPyNet | 2.14 FlowFormer | FlowFormer |
| 6 | 5.64 RAFT | 2.60 RAFT | SPyNet |
| 7 | 7.01 FlowNet2 | 2.77 PWCNet | PWCNet |
| 8 | 7.25 PWCNet | 2.82 SPyNet | RAFT |

flow across corruptions with a lower average $R^c_{\text{EPE}}$ than RAFT-3D. But both methods suffer significant performance losses for severe weather like rain and noise-based corruptions, *e.g.* impulse noise. Interestingly, their robustness does not improve compared to conventional optical flow models. Noise and weather corruptions remain a challenge for Disparity 2 predictions. Here, RAFT-3D consistently achieves lower robustness scores compared to M-FUSE, but conditions like impulse noise or rain still notably affect disparity predictions. Overall, both models have limited robustness, but temporal consistency may contribute to lower robustness scores under several corruption types.

**Stereo.** The results of the stereo disparity estimations are presented in Tab. 2b. The effect of the different corruptions on the performance is significant, with noise and weather-based corruptions leading to the largest errors, especially for GANet and LEAStereo. In particular, Gaussian and impulse noise introduce extremely large errors, highlighting the sensitivity of stereo models to pixel-level noise. Blur distortions, especially zoom blur, also have a severe impact on all models, with high 1px and D1 errors. In contrast, color-based distortions generally yield smaller errors. RAFT-Stereo shows stronger resilience across most corruption groups, performing better on color and noise based corruption than other models. However, it also struggles with noise and severe weather effects such as rain and snow.

## 4.2 Metrics and Benchmark Capability

After reporting initial RobustSpring results, we analyze aspects of its benchmark character: The subsampling strategy for data efficiency, and different ranking systems for result comparisons across 20 different prompt variations. We also validate our robustness metric for object corruptions and explore RobustSpring's transferability to the real-world.

**Subsampling.** We evaluate RobustSpring's strict data subsampling by comparing to results on the full test set. As shown in Tab. 3a, our subsampling strategy produces results that are nearly identical to those that include all pixels in the robustness calculation. We observe the largest discrepancy for Spring's original subsampling, because it includes a handful of full-resolution Hero-frames. If those frames are also subsampled (Spring*), results align with the full dataset. Overall, our stricter subsampling to 0.05% of all data is not only data efficient but also exact.

**Metric Ranking.** To explore how ranking strategies influences the optical-flow robustness order, we contrast our three summarization strategies: Average, Median, and Schulze, *c.f.* supplement. The rankings in Tab. 3b notably differ across strategies. The Average differs most from the other rankings. For example, it ranks GMFlow 1st, which is only 4th on Median and Schulze, suggesting a good performance across corruptions without excessive outliers but no top performance on most corruptions. Interestingly, Median and Schulze rankings are more aligned. As Schulze's ranking involves complex comparisons of per-corruption rankings and must be globally recomputed for new models, the Median ranking is a cheap approximation to it. The ranking strategy has significant implications for selecting robust models. No model is optimal across rankings, and the rankings accentuate different aspects: overall performance, outlier robustness, or balanced performance in pairwise comparisons. Hence, RobustSpring reports them all.

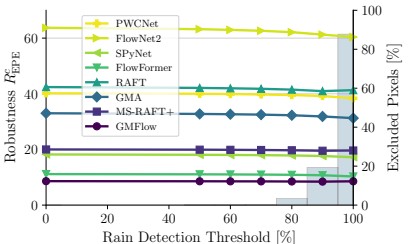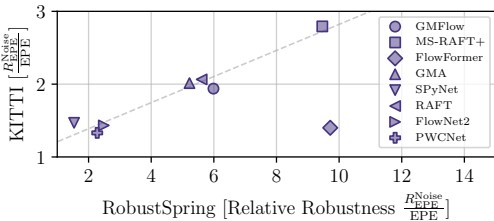

(a) Stability of corruption robustness $R_{\mathrm{EPE}}^c$ on rain corruption. Robustness scores and rankings remain stable even if no rain pixels are in the $R_{\mathrm{EPE}}^c$ calculation.

(b) Relative robustness to noise on RobustSpring transfers to noisy real-world KITTI data [41] for most optical flow models.

Figure 5: Additional evaluations of RobustSpring's benchmark character.

**Corruption Robustness on Object Corruptions.** Intuitively, models are robust if they recover the main scene despite image corruptions. Here, we investigate if the corruption robustness metric faithfully represents model robustness even if corruptions like rain introduce moving objects to the scene. To this end, we contrast the robustness score contributions of background and corruption objects, by excluding pixels of objects like rain drops from the score calculation. We detect object pixels by taking the value difference $d$ between original and corrupt images, and exclude them if $(1-d)$ is above a detection threshold. Threshold 0 detects no rain pixels, matching the vanilla $R_{\mathrm{EPE}}^{\mathrm{Rain}}$, while 100 detects all. Figure 5a shows the robustness score if rain is excluded from the calculation, along with bars indicating the amount [%] of excluded pixels. Remarkably, the robustness score is stable, *i.e.* varies $\leq 5\%$, even for discarding *all* rain pixels, *i.e.* 90% of all pixels. Large robustness scores on rain or snow, *c.f.* supplement, thus stem from mispredictions in the *periphery* of altered pixels, not from motion predictions on altered pixels. As scene-wide effects dominate it, our corruption robustness yields stable robustness rankings that make it suited for broad model robustness evaluations.

**Robustness in the Real World.** Finally, we investigate if RobustSpring's corruption robustness transfers to the real world. To this end, we select the noisiest 10% KITTI data, estimating noise as in [21]. These noisy KITTI frames have no clean counterparts to calculate corruption robustness $R_{\mathrm{EPE}}^{\mathrm{Noise}}$. Thus, we approximate $R_{\mathrm{EPE}}^{\mathrm{Noise}}$ via the accuracy difference on noisy and non-noisy KITTI frames. To account for model-specific performance differences on Spring and KITTI, we normalize with the clean dataset performance and show the resulting relative robustness $\frac{R_{\mathrm{EPE}}^{\mathrm{Noise}}}{\mathrm{EPE}^{\mathrm{Clean}}}$ in Fig. 5b. Relatively robust models with low scores on RobustSpring are also robust on KITTI and vice versa. The only outlier, FlowFormer, overperforms on KITTI, potentially due to outstanding memorization capacity and exposure to KITTI during training. Because overall noise resilience on RobustSpring qualitatively transfers to KITTI, RobustSpring supports model selection for real-world settings where corruption robustness cannot be measured.

## 5 Conclusion

With RobustSpring we introduce an image corruption dataset and benchmark that evaluates the robustness of optical flow, scene flow and stereo models. We carefully design 20 different image corruptions and integrate them in time, stereo, and depth for a holistic evaluation of dense matching tasks. Furthermore, we establish a corruption robustness metric using clean and corrupted predictions, and compare ranking strategies to unify model results across all 20 corruptions. RobustSpring's benchmark further supports data-efficient result uploads to a public website. Our initial evaluation of 16 optical flow, scene flow and stereo models reveals an overall high sensitivity to corrupted images. As our robustness results translate to real-world performance, systematic corruption benchmarks like RobustSpring are crucial to uncover potential model performance improvements.

**Limitations.** Due to its benchmark character, we have limited the image corruptions on RobustSpring to a selection of 20. While this does not cover the full space of potential corruptions, this data-budget limitation is necessary to make the RobustSpring dataset applicable and not overburden the computational resources of researchers during evaluation.

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
