# OpenReview forum: "RobustSpring: Benchmarking Robustness to Image Corruptions ring: Benchmarking Robustness to Image Corruptions for Optical Flow, Scene Flow and Stereo"
_NeurIPS.cc/2025/Datasets_and_Benchmarks_Track — Submitted to NeurIPS 2025 Datasets and Benchmarks Track_

### Official Review · Reviewer_npBi · 2025-06-05

**Rating:** 6
**Confidence:** 4

**Summary:**

Overall a very good paper, with only minor corrections required (I hesitate between a 5 and a 6).

**Additional Feedback:**

The kind of noise described the the abstract: ("noise or rain") are inconsistent with the kind described in the introduction ("sensor noise or compression artifacts"). I feel that the version in the abstract is more in line with the kind of degradations used in the paper.

There is a typo on line 20 on page 2: "naviation" is written instead of "navigation".

Line 36 on page 2 "monocular 2D or 3D space" and line 76 on page 3 "2D image corruptions": I think it would be clearer to state that the noise is CONSISTENT only on a single 2D image or an image pair (but inconsistent in time).

The authors missed a few references related to robustness in stereo matching and optical flow that could be added:

Maddern, W., Pascoe, G., Linegar, C., & Newman, P. (2017). 1 year, 1000 km: The oxford robotcar dataset. The International Journal of Robotics Research, 36(1), 3-15.

Diaz-Ruiz, Carlos A., et al. "Ithaca365: Dataset and driving perception under repeated and challenging weather conditions." Proceedings of the IEEE/CVF Conference on Computer Vision and Pattern Recognition. 2022.

Jospin, L., Antony, A., Xu, L., Laga, H., Boussaid, F., & Bennamoun, M. (2022). Active-passive simstereo-benchmarking the cross-generalization capabilities of deep learning-based stereo methods. Advances in Neural Information Processing Systems, 35, 29235-29247.

**Dataset Code Accessibility:**

Yes

**Dataset Code Comments:**

An online tool is available at https://spring-benchmark.org/download, with the dataset and the interface to submit to the benchmark.

**Ethical Considerations:**

No, there are no or only very minor ethics concerns

**Final Justification:**

So, after reading the authors rebuttal, engaging in discussion with the authors and checking the comments and following rebuttal and discussion (or lack of discussion) from the other reviewers, I have decided to keep my rating and recommend acceptance.

I feel that the reviewers agree, or at least do not disagree, that the topic is important and the paper well written.

Reviewer AZjQ raised two major concerns: that optical flow is not defined and that the validity of the ground truth after certain corruptions is in question.
In my opinion, while the two concerns were valid they are trivial to fix as the optical flow definition is a matter of adding two sentences and an equation, and/or giving a proper reference. The authors provided both in their rebuttal. For the validity of the ground truth after corruption, the authors argument that since their benchmark is more concerned about robustness than accuracy, and based on their definition of optical flow, it is not an issue to consider that the corruption and ground truth should be dissociated. I feel that the argument comes down to intended use of the dataset. Models trained for autonomous driving obviously need to be robust to snow/rain being present in the scene and/or optical distortions caused by water on the cameras. Models planned to be used for video editing on the other hand will have to treat the rain/snow not as noise but an integral part of the scene. In general certain categories of noise will have limitations that just need to be made explicit in the final manuscript. As presented in table 1, results should be available per corruption type, so end-users can ignore one specific kind of corruption if it is not relevant/should not be considered as noise for their intended application. The reviewer did not engage in the discussion, so I cannot judge if the reviewer would be satisfied with the changes, but the authors described the intended changes in a fashion I find convincing.

Reviewer z5SM raised the issue of limit in the range of motions, kind of deformation and objects interactions and domain gap for the robustspring benchmark. The authors have addressed these concerns by pointing out that the spring dataset, upon which the robustspring benchmark is based, already offer improvements compared to previous state of the art benchmark, and agreed to discuss some of the points raised by the reviewer in the limitation section. In my opinion the answers mostly addresses the issues raised by the reviewer, and the reviewer already had rated the paper as a borderline accept.

Reviewer xXnY raised issues related to the COLMAP based depth for corruption (which I have also raised), lack of abblation study for attribution analysis and issues related to validation on real scenes and related to occlusion and motion boundary. The reviewer have answered to the authors rebuttal and indicated that the answers and proposed changes have addressed their concerns.

As I have indicated below, my concerns have also been addressed.

As stated above, I feel the reason to recommend acceptance clearly outweighs the reasons to reject and thus have kept my initial rating.

**Limitations Weaknesses:**

The data is simulated, which is both a curse and a blessing. I would argue if the authors were using some specific benefits of simulation that would not be an issue, but there are a few parts in the paper where the authors described their processing of the data just as if they did not have access to the ground truth at all (especially line 101 on page 3, where the authors describe using COLMAP to estimate extrinsic parameters). I would appreciate if the authors in their rebuttal can clarify why this was necessary (I mean, the team is quite similar to the spring dataset, so I would guess even if it is not public, the authors have access to the original dataset ground truth and to the 3d files used to produce the dataset).

Another limitation is that the authors did not evaluate how their dataset can be used for finetuning, and what is the impact of finetuning with the proposed dataset. While I agree that this is not 100% required for a robustness benchmark, this is still pretty common to do it for research works in computer vision. I wonder if the authors could provide the results for at least one of the method after it being finetuned on the proposed dataset.

**Strengths Contributions:**

This work propose a robustness benchmark for dense matching tasks (scene flow, optical flow, dense stereo). Based on the spring dataset, itself based on the assets of the spring open movie, the dataset offer a good amount of details in the images and a large amount of data.

Robustness is important for the tasks at hand, as most deep learning models for scene flow, optical flow and dense stereo are used for autonomous driving (not only, but this is one of the main application). As such, robustness is of utmost importance for the security of the users.

Good quality sceneflow and optical flow datasets are harder to come by than stereo datasets.

An online benchmarking tool with withholded ground truth exist, which makes it fair and convenient to use the proposed benchmark has been implemented and is already available.

---

> ### Author Rebuttal · Authors · 2025-07-31
>
> Thank you for your thoughtful review and comments, and your very positive evaluation of our RobustSpring benchmark!
> Below, we would like to address your two questions regarding depth estimation and model finetuning.
>
> It is correct that we did not use ground truth data even though we have access to it. The reason is that for some of the augmentations (particularly fog, with the Koschmieder model that adds more color to deeper pixels), it would become quite easy to get accurate depth estimates by taking the pixel-wise difference between the original and the fog-augmented frame. Therefore, using the exact depth would leak some parts of the ground truth and endanger the integrity of our original Spring benchmark. We agree that using the estimated depths is not ideal, but our priority was to maintain the integrity of the original benchmark while adding a corruption benchmark. The depth estimates are only necessary for a limited number of corruptions, and which sacrifices a small amount of realism for 3/20 corruptions but maintains the original (uncorrupted) Spring benchmark.
>
> In response to Reviewer xXnY we also quantified the error that is introduced by our estimation (MS-RAFT+ disparity estimation, mean EPE = 1.09 px; D1-all = 5.60% [percentage of pixels with error larger than 3px or 5% of ground truth disparity]), which is an acceptable value given that the depth estimates are only used for 3/20 corruptions (motion blur uses depth implicitly rather than explicitly).
>
> Also, we agree that it would be interesting to report results after finetuning to corrupted images in RobustSpring. Unfortunately this type of experiment is not supported by the current dataset and exceeds our computational resources to run in one week. RobustSpring only provides corrupted versions of the test proportion of the Spring data, and therefore does not come with ground truth annotations that would facilitate supervised model finetuning. These annotations are available in the Spring training proportion, which we purposefully did not augment to discourage the use of our augmentations as training augmentations. Only augmenting the test proportion of images already took two days for some of the scene-consistent particle effects snow and rain. Therefore, augmenting the substantially larger training proportion of Spring with these effects exceeds one week, not even including model training. On a broader note, we did not add an experiment like this to the original version of the paper, as we explicitly discourage the use of RobustSpring's corruptions for fine-tuning (L.189 of the paper). We felt that it would be diametric to our discouragement to then design an experiment that does exactly this, but generally agree that a quantification of this effect would have been useful.
>
> Also, thank you for your great care in reading the paper. We have clarified the manuscript according to your suggestions (unified wording in abstract and introduction, clarified noise consistency) and added the related publications to the revised version of our paper.

---

> > ### Comment · Reviewer_npBi · 2025-08-02
> > **Thanks for your answer**
> >
> > Thanks for answering my comments. I have read your answer, as well as the answers to the other reviewers comments. Overall, even if I was more enthusiast than the other reviewers, I feel that it is warranted and will not lower my score.
> >
> > Sorry for my misunderstanding about the use of COLMAP. I did not understood the sentence "however, we require the extrinsic camera parameters and depths that are withheld", in the camera ready version it could be interesting to add " to preserve the integrity of our original Spring benchmark" just for clarity.
> >
> > I understand that some corrections cannot be made in the short amount of time given to write the rebuttal.

---

> > > ### Author Response · Authors · 2025-08-05
> > >
> > > Thank you for your reply, and we are happy that the rebuttal answered the remaining questions. We will include the suggested note about the reasons for withholding extrinsics and depth in the camera-ready.

---

> > ### Comment · Area_Chair_Hc9j · 2025-08-05
> >
> > Dear Reviewers
> >
> > Thanks for contributing to Neurips. We have received new information from the PCs:
> > “Reviewers must participate in discussions with authors before submitting “Mandatory Acknowledgement”. ” “To facilitate discussions, we extend Author-Reviewer discussions by 48h till Aug 8, 11.59pm AoE. ”
> >
> > As informed, please engage in discussions with the authors.
> >
> > Kind regards
> > AC

---

### Official Review · Reviewer_xXnY · 2025-06-23

**Rating:** 4
**Confidence:** 5

**Summary:**

The paper proposes RobustSpring, the first image corruption robustness benchmark for optical flow, scene flow, and stereo vision tasks. Based on the high-resolution Spring dataset, a 40,000-frame evaluation dataset is constructed by introducing 20 types of temporal/stereo/depth consistent image corruptions (such as noise, blur, rain and snow).

**Dataset Code Accessibility:**

Yes

**Dataset Code Comments:**

In this paper, the related link has released the dataset and code, and the reviewer can have access to obtain it.

**Ethical Considerations:**

No, there are no or only very minor ethics concerns

**Final Justification:**

The authors have addressed my concerns.

**Limitations Weaknesses:**

(1) Depth consistency relies on estimation, which introduces potential errors. Since the Spring test set does not provide the true depth value, the camera's external participation depth map relies on COLMAP and MS-RAFT+ estimation. The impact of estimation errors on depth consistency damage is not quantified.

(2) The attribution analysis of model robustness is insufficient. It only mentions that Transformer global matching may lead to noise sensitivity, but it is not verified by ablation experiments.

(3) Real scene verification limitations: Real scene migration only verifies noise (KITTI subset) and does not cover key weather damage such as rain and fog.

(4) The evaluation dimensions do not cover key challenges. The robustness to occluded regions and motion boundaries is not evaluated, which are common failure points for optical flow/stereo tasks.

**Strengths Contributions:**

(1) The problem analysis is thorough. It accurately points out the defects of existing benchmarks (such as KITTI and Sintel) that over-focus on accuracy and ignore robustness evaluation.

(2) The method is novel in design. It breaks through the limitations of traditional 2D corruption and pioneers temporal, stereoscopic, and depth consistency corruption, which closely fits the characteristics of optical flow/stereoscopic tasks.

---

> ### Author Rebuttal · Authors · 2025-07-31
>
> We thank you for your time and the positive review of our work!
> Below we would like to address some of the mentioned concerns.
>
>   * (1): It is correct that our depth estimation introduces potential errors. We have quantified the error that is introduced by our depth estimation with COLMAP and MS-RAFT+:
>     * (a) MS-RAFT+ disparity estimation: Mean EPE = 1.09 px, D1-all = 5.60%. Mean-EPE quantifies the average Euclidean distance between the predicted and ground truth disparities. D1-all quantifies the percentage of pixels where the disparity error is larger than 3 pixels or 5% of the ground truth disparity. Overall, these errors are small enough to provide meaningful augmentations for the 3/20 corruption types that use the depth estimates (motion blur uses it only implicitly). The estimation itself is necessary to avoid ground truth leakage, and therefore is our solution to the trade-off between protecting the ground truth data and generating realistic augmentations.
>     * (b) COLMAP camera extrinsics estimation: The trajectories are not expressed in a common reference frame, and without a rigid alignment step that would itself introduce errors, quantitative metrics like absolute trajectory error (ATE) or relative pose error (RPE) cannot be meaningfully computed. The extrinsics are used by 2/20 corruptions (snow, rain). Therefore, we can only assess results qualitatively (e.g. via the provided video in the supplement), which shows a realistic motion behavior for both effects.
>   We have added these quantitative results to a revised version of our paper.
>
>   * (2) We agree that additional ablation experiments are interesting. One of the merits of this new benchmark is that it provides the base for authors to run these ablation experiments.
>
>   * (3) Real scene verification experiments are hard to design due to a lack of annotated real-scene data, especially for weather conditions like rain and snow. To the best of our knowledge, KITTI is the only real-world dataset that has annotations for optical flow and scene flow, from which we extracted noisy scenes. Most available real-world datasets with weather include no optical flow and no scene flow annotations, and very few have LiDAR depth [A,B] for disparity estimation. We would be happy to extend our experiments if there exist suitable datasets for optical flow or scene flow that we are unaware of. However, our noise experiment, together with existing studies on image segmentation (which indicate that synthetic corruptions are reliable approximations to real-world corruptions [C]), is evidence that synthetic evaluations are suited to approximate the real world. We have extended the discussion with additional literature on the synthetic-to-real domain gap for image corruption evaluation in a revised version of the paper.
>
>   * (4) These key challenges are already quantified in the original Spring benchmark, which measures the quality in occluded areas and areas with high detail (motion boundaries). However, we conducted an additional analysis where we use the occlusion maps from Spring and apply them to the evaluation of RobustSpring (see Table below). This further demonstrates the flexibility of our benchmark, as it becomes possible to apply the area-specific evaluations (occlusions, detail, sky, rigid motion) from Spring on top of our corruption benchmark:
>
> Comparing Robustness in occluded regions (occ) to the whole flow (all) for MS-RAFT+
>
> | Category    | Distortion | **occ** | **occ** | **occ** | **all** | **all** | **all** |
> |-------------|------------|------------------|-------|-------|--------------|-------|-------|
> |             |            | Rc_EPE | Rc_1px | Rc_Fl | Rc_EPE | Rc_1px | Rc_Fl |
> | **Color**   | Brightness | 3.83 | 23.27 | 10.92 | 0.33 | 2.88 | 1.02 |
> |             | Contrast   | 7.66 | 34.08 | 18.68 | 0.87 | 6.69 | 3.24 |
> |             | Saturate   | 4.01 | 23.27 | 10.79 | 0.34 | 2.87 | 1.03 |
> | **Blur**    | Defocus    | 5.57 | 29.31 | 14.49 | 0.51 | 4.01 | 1.47 |
> |             | Gaussian   | 5.98 | 30.63 | 15.41 | 0.58 | 4.45 | 1.63 |
> |             | Glass      | 5.16 | 30.63 | 13.43 | 0.53 | 4.45 | 1.37 |
> |             | Motion     | 9.41 | 50.51 | 32.10 | 1.31 | 14.06 | 6.16 |
> |             | Zoom       | 12.38 | 57.60 | 32.04 | 1.81 | 21.84 | 7.13 |
> | **Noise**   | Gaussian   | 19.82 | 70.15 | 50.65 | 5.70 | 35.74 | 22.12 |
> |             | Impulse    | 22.39 | 79.36 | 59.03 | 7.39 | 45.72 | 29.05 |
> |             | Speckle    | 16.67 | 71.79 | 47.92 | 4.22 | 34.96 | 17.18 |
> |             | Shot       | 17.09 | 67.55 | 46.67 | 4.36 | 31.67 | 17.77 |
> | **Quality** | Pixelate   | 8.61 | 70.04 | 24.47 | 1.60 | 45.83 | 6.78 |
> |             | JPEG       | 11.67 | 74.30 | 39.12 | 2.09 | 41.69 | 12.82 |
> |             | Elastic    | 5.07 | 61.56 | 21.98 | 1.16 | 32.49 | 5.54 |
> | **Weather** | Fog        | 4.90 | 27.03 | 14.65 | 0.91 | 10.32 | 6.33 |
> |             | Frost      | 31.71 | 62.16 | 52.81 | 7.38 | 29.96 | 21.25 |
> |             | Rain       | 39.06 | 64.00 | 51.60 | 19.99 | 36.74 | 31.22 |
> |             | Snow       | 15.69 | 57.90 | 47.74 | 4.69 | 33.21 | 30.91 |
> |             | Spatter    | 29.92 | 62.04 | 52.16 | 6.63 | 28.22 | 20.24 |
> | **Average** |            | **13.83** | **52.36** | **32.83** | **3.62** | **23.39** | **12.21** |
> | Std. Dev.   |            | 10.25 | 19.28 | 17.03 | 4.58 | 15.54 | 10.62 |
> | **Median**  |            | **10.54** | **59.73** | **32.07** | **1.71** | **29.09** | **6.95** |
>
>
>
> [A] Diaz-Ruiz, Carlos A., et al. Ithaca365: Dataset and driving perception under repeated and challenging weather conditions. Proceedings of the IEEE/CVF Conference on Computer Vision and Pattern Recognition. 2022.
> [B] Maddern, W., Pascoe, G., Linegar, C., & Newman, P. 1 year, 1000 km: The oxford robotcar dataset. The International Journal of Robotics Research, 2017.
> [C] Agnihotri, Schader, Sharei, Kaçar and Keuper. Are synthetic corruptions a reliable proxy for real-world corruptions? CVPR'25 Workshop on Synthetic Data for Computer Vision.

---

> > ### Comment · Area_Chair_Hc9j · 2025-08-05
> >
> > Dear Reviewers
> >
> > Thanks for contributing to Neurips. We have received new information from the PCs:
> > “Reviewers must participate in discussions with authors before submitting “Mandatory Acknowledgement”. ” “To facilitate discussions, we extend Author-Reviewer discussions by 48h till Aug 8, 11.59pm AoE. ”
> >
> > As informed, please engage in discussions with the authors.
> >
> > Kind regards
> > AC

---

> > ### Comment · Reviewer_xXnY · 2025-08-06
> >
> > After carefully reviewing the rebuttal, my concerns have been addressed. I am happy to maintain my original score and recommend acceptance. I look forward to seeing the final version of the paper.

---

### Official Review · Reviewer_z5SM · 2025-07-02

**Rating:** 4
**Confidence:** 4

**Summary:**

This paper introduces RobustSpring, the first systematic image‐corruption benchmark tailored to dense matching tasks. Building on the high-resolution Spring dataset, the authors apply 20 diverse corruptions and integrate them in time, stereo, and depth for a holistic evaluation of dense matching tasks. They propose a ground-truth-free robustness metric based on Lipschitz continuity to disentangle robustness from accuracy, and integrate these evaluations into Spring’s public benchmark site. As their robustness results translate to real-world performance, systematic corruption benchmarks like RobustSpring are crucial to uncover potential model performance improvements.

**Additional Feedback:**

The dataset for this paper is an improved version of an existing one, and while there are plenty of improvements, the originality is always weaker.

**Dataset Code Accessibility:**

Yes

**Dataset Code Comments:**

This paper provides the reproducing lines " Dataset URL: https://huggingface.co/datasets/jeschmalfuss/RobustSpring, and
Code URL: https://github.com/cv-stuttgart/springwebsite/tree/main/subsampling", but lack of supplementary material and less detail of reproduction.

**Ethical Considerations:**

No, there are no or only very minor ethics concerns

**Final Justification:**

I agree with the above reviewer: the problem is not that the authors “break” the definition of optical flow, but that the benchmark scope is under-specified. Treating snow/rain/dust as transient noise is a valid hypothesis if the task is “flow of stable scene surfaces only.” Please make this explicit and align the evaluation to that scope.

**Limitations Weaknesses:**

1. Frame-to-frame motions are on the order of 0.01–0.02 s (only a few pixels of shift), offering no challenge for large-displacement flow estimation.
2. Fully synthetic environments lack real-world lighting variations, complex occlusions, and material properties—introducing a substantial domain gap.
3. The tiny motion range prevents models from learning to handle large-scale or non-rigid movements common in real scenes.
4. Few multi-object interactions, deformations, or rapid accelerations, so the dataset cannot assess robustness to complex, nonlinear motion patterns.

**Strengths Contributions:**

1. Provides perfectly aligned color (RGB), 2D optical flow, and 3D scene flow annotations in one dataset, enabling joint evaluation of single-view and multi-view, 2D and 3D matching methods.
2.  Fully synthetic data gives exact, error-free ground truth—ideal for benchmarking upper-bound performance.
3. This paper proposes a corruption robustness metric, based on Lipschitz continuity, which subsamples the clean-corrupted prediction difference and disentangles robustness and accuracy.

---

> ### Author Rebuttal · Authors · 2025-07-31
>
> We appreciate the time you took to review our work and the positive evaluation of RobustSpring! Regarding the mentioned weaknesses, we would like to clarify a few points, especially how the original Spring benchmark is related to RobustSpring.
>
> It is mentioned in the review that RobustSpring is an improved version of an existing paper. We assume this existing paper refers to Spring? While it is correct that the RobustSpring Corruption dataset builds upon the Spring dataset, we would like to highlight that RobustSpring is its own robustness benchmark, with tailored corruptions, a newly developed robustness metric, and careful evaluations to demonstrate the benchmark's use. Though we designed RobustSpring to be a complementary benchmark to Spring, its value is not tied to the Spring benchmark, and it can function and stand on its own.
>
>
>   * (1)+(3): Because our contributions (time-, depth- and stereo-dependent corruptions; corruption robustness metric) may be applied to any optical flow, stereo or disparity estimation dataset, several of the mentioned weaknesses seem to be directed to the underlying Spring data rather than our RobustSpring dataset. If our read of them is correct, we would like to add more perspective on the motion ranges in the original Spring data: We purposefully based our robustness benchmark on Spring because it covers more complex and larger motions than standard benchmarks like KITTI or Sintel. In the original Spring publication, its motion ranges are compared to the Sintel and KITTI benchmarks in Fig. 5 [Spring paper]: While the motion statistics are aligned over all three benchmarks for smaller motions (u:-500 to 500, v: -250 to 250), Spring offers ground truth motion with significantly larger motion range of up to 1700 (u) and -750 (v), while Sintel and KITTI's maximal motions are only 1/3 to 2/3 of the maximal motion length in Spring. Compared to Sintel and KITTI, Spring thus offers significantly larger motion ranges. We have added a comment on Spring's motion statistics in a revised version of the paper
>
>   * (2): Regarding our RobustSpring benchmark, we agree that using fully synthetic data introduces a domain gap. However, during model training, models are not tied to using synthetic data only (which has been shown to decrease corruption robustness compared to mixed real-world data training [A]). Furthermore, in the context of our benchmark itself, prior research on semantic segmentation models indicates that synthetic corruptions can serve as a proxy for real-world corruptions, and that model performance evaluation on synthetic corruptions approximates model performance on real data [B]. We make similar observations for optical flow models when we investigate how noise corruption on RobustSpring transfers to robustness to noise on the real-world KITTI dataset in Fig. 5b, which indicates that the domain gap is small.
>
>   * (4): Regarding multi-object interactions, deformations and rapid accelerations, it would be interesting to know if a specific type is requested. We acknowledge that rapid accelerations are currently not represented in the RobustSpring data. However, multi-object interactions are introduced with the snow and rain corruptions, where the weather particles interact among themselves and with the background objects that are present in the Spring data. With RobustSpring, we even demonstrate in the rain experiment in Fig. 5a that multi-object interactions pose a robustness problem to all models, as the majority of prediction errors come from the background motion rather than the motion of small, fast-moving objects themselves. Furthermore, image deformations are introduced with the elastic transform corruption. We will explicitly mention the lack of rapid accelerations in the limitations. Given that RobustSpring includes multi-object interactions and deformations it would be interesting to know if we are missing specific corruptions that better represent those types.
>
>
> [A] Singh, Navaratnam, Holmer, Schaub-Meyer, Roth. Is synthetic data all we need? Benchmarking the robustness of models trained with synthetic images. CVPR'24 Workshops (CVPR-W)
> [B] Agnihotri, Schader, Sharei, Kaçar and Keuper. Are synthetic corruptions a reliable proxy for real-world corruptions? CVPR'25 Workshop on Synthetic Data for Computer Vision.

---

> > ### Comment · Area_Chair_Hc9j · 2025-08-04
> >
> > Dear Reviewer
> >
> > The Author-Reviewer discussion phase is until 6 Aug. The author has submitted a rebuttal. Please feel free to initialize discussions with the authors.
> >
> > Kind regards
> > AC

---

> > ### Comment · Area_Chair_Hc9j · 2025-08-05
> >
> > Dear Reviewers
> >
> > Thanks for contributing to Neurips. We have received new information from the PCs:
> > “Reviewers must participate in discussions with authors before submitting “Mandatory Acknowledgement”. ” “To facilitate discussions, we extend Author-Reviewer discussions by 48h till Aug 8, 11.59pm AoE. ”
> >
> > As informed, please engage in discussions with the authors.
> >
> > Kind regards
> > AC

---

### Official Review · Reviewer_AZjQ · 2025-07-02

**Rating:** 2
**Confidence:** 4

**Summary:**

The paper presents an extension of the established Spring dataset/benchmark for estimating the robustness of correspondence approaches. In particular, the authors present RobustSpring for evaluating the robustness of optical flow, stereo, and scene flow approaches to image corruptions. A new variant of Spring - RobustSpring - including relevant image corrections, is proposed. To validate robustness without ground-truth, a robust evaluation metric is presented. The authors provide initial robustness results and analyze their novel benchmark/dataset.

**Additional Feedback:**

Currently, I need to rate the paper as reject, due to an unclear definition of optical flow and significant doubts about the validity of the robust evaluation. However, I still believe the general idea of the paper is novel, and an evaluation on corrupted images is needed to make progress in optical flow, disparity, and scene flow estimation. I'm looking forward to the response of the authors, and I'm happy to significantly raise my score if the authors can address my concerns, as, apart from my two main concerns, the paper is well written and relevant.

Minor comments:
- It might be beneficial to not only present initial results using supervised approaches but also for unsupervised/self-supervised models, such as SMURF [Stone et al., 2021].
- Fig. 2a elastic transform is not fully displayed.
- The used JPEG compression strength seems a bit strong, as in practice, weaker compression rates are used; still, the SSIM motivation is clear
- The intended use is to use RobustSpring just as a validation dataset; however, what convinces people to just use the presented corruptions as data augmentations during training on Spring (train). This could lead to cheating of the benchmark.
- Finally, I feel the initial results are missing a discussion about data augmentation during model training, as not all models use the same data augmentation pipeline and potentially include augmentations similar to the presented image corruptions.

**Dataset Code Accessibility:**

Yes

**Dataset Code Comments:**

The presented dataset and benchmark suite are accessible, well-documented, and in a usable format. The only small inconvenience is that each corruption needs to be downloaded separately.

**Ethical Comments:**

The presented dataset is synthetic and validates low-level vision concepts, thus not posing any ethical concerns.

**Ethical Considerations:**

No, there are no or only very minor ethics concerns

**Final Justification:**

As discussed in detail below, I'm divident about the paper. The paper is very well written, the validation approach seems valid for current models and training pipelines, while not the first paper to present a robustness benchmark in this domain, it's the most comprehensive, and validating robustness for dense matching is relevant. The integration into the existing Spring dataset is also convenient. Still, I have doubts about the validity of the benchmark (see comments), and subsequently, the potential negative impact the benchmark might have on the community. I'm not confident in assessing whether the potential positive impact outweighs the potential long-term negative impact by not providing a meaningful lower bound of the robustness score, and therefore, I maintain my current score. **That said, I strongly encourage the AC to carefully assess my concern regarding the benchmark's validity and potential negative impact. If these concerns are deemed unfounded and the positive impact is judged to outweigh the negatives, please downweight my review accordingly in the final decision and consider accepting the paper!** Finally, I would like to thank the authors for their rebuttal and reviewer npBi for his comments on my points.

**Limitations Weaknesses:**

In general, there are two major limitations of the presented benchmark. First, a clear *definition of optical flow*, and second, doubts about the *validity of the overall validation*.

The paper does not clearly define how optical flow is defined. Most datasets define optical flow as the "true" motion field, i.e., the true 3D motion of surfaces projected into 2D [Horn, 1986] [Baker et al., 2011]. While other papers follow the definition of apparent motion. This is a limitation and has significant implications for the presented validation. This point also applies to the definition of scene flow estimation.

The authors present a ground-truth-free evaluation approach for measuring the robustness of optical flow, stereo, and scene flow. While this approach has benefits in terms of practicality, it introduces significant ambiguities and doubts about the validity. While image captions like brightness and contrast changes do not change the motion field and (in most cases) the apparent motion, other corruptions do. Corruptions like JPEG coding might not change the motion field, but definitely affect the apparent motion, due to pixel-wise discretization of pixels. Similarly, motion blur leads to ambiguities w.r.t. the apparent motion in boundary areas. Even more significant corruptions like snow, rain, and elastic transform change both the motion field and the apparent motion. Measuring robustness now as the difference between the clean and corrupted predictions is ambiguous and seems invalid. For example, in the ideal case, an optical flow approach, modeling the motion field or apparent motion, would predict motion vectors for snowflakes and raindrops. Additionally, when elastically transforming the input frames, the optical flow would also capture the difference between each transform. Just naively assuming that the approach should ideally estimate the motion field of the uncorrupted frames for the corrupted frames seems not valid.
While one could argue that the presented ground-truth-free evaluation is an approximation of validating with the actual ground truth, the authors miss a thorough discussion on this and should provide a clear validation with ground truth to showcase for which corruptions and model accuracy their evaluation is a valid approximation. However, in the current version of the paper, I'm not convinced by the presented ground-truth-free evaluation.

**Strengths Contributions:**

- The presented benchmark approaches a highly relevant and interesting problem, which is not fully captured/evaluated by existing benchmarks (e.g., Sintel).
- The image corruptions are well motivated and relevant.
- The presented subsampling makes benchmarking efficient as well as accessible and is well justified.
- The writing and general presentation of the paper are clear and mostly easy to follow
- Initial evaluation is good while not fully complete (cf. weaknesses)

---

> ### Author Rebuttal · Authors · 2025-07-31
>
> We would like to thank you for your in-depth review of our paper. Based on your questions, we believe that the concerns about definitions and evaluation validity are a misunderstanding that originates from the brevity of explanations on RobustSpring's robustness definition. RobustSpring's robustness definition is based on the understanding that robustness and accuracy are distinct (and potentially contradicting) model properties [56,67,64,16,45], where the definition of the optical flow (and thus your question what is considered the ground truth) only affects model accuracy, but not the robustness and its measurement.
>
> We start with model accuracy, which measures the difference between the model prediction and a known ground truth and thus is influenced by the definition of optical flow. For accuracy, RobustSpring inherits the optical flow definition of the original Spring benchmark, which defines it as true 3D motion of surfaces projected into 2D [Horn, 1986] [Baker et al., 2011]. For clarity, we now explicitly mention this definition in the revision of the paper. If we were to measure method accuracy on the corrupted RobustSpring scenes, it would indeed matter that elastic transforms, snow, rain, or even blurs change the motion boundaries of existing objects and would require identifying _how_ these changes affect the ground truth flow field. However, we do not need to measure this difference because we use clearly separated definitions for accuracy and robustness in RobustSpring.
>
> For model robustness, we rely on the established concept of the Lipschitz constant (Eq. 1) which measures how much the model prediction changes in relation to the input change (the corruption strength). Consequently, a model has ideal robustness if it always makes the same prediction - independent of the input.
>
> While we acknowledge that ideally Lipschitz-robust models are of little use for practical applications, this definition of robustness is still useful for the real world, because it rewards models that make stable scene predictions _despite_ the presence of small perturbances like noises, blurs or weather. Furthermore, this definition is established in the robustness research community [56,67,64,16,45], as it provides an additional axis for model evaluations that complements accuracy. With both quantities, practitioners can choose if more accurate or more stable predictions are required, and adjust this trade-off depending on their use case.
>
> Few works [4,49] propose to measure robustness as the difference between the prediction on corrupted images and the ground-truth (rather than the clean prediction). However, this approach entangles robustness and accuracy and consequently sparks all the questions in the review with regard to how much each corruption changes the original flow field. It is also this ground-truth-dependent robustness definition (not the one we use for RobustSpring), which assumes that robust methods should estimate the motion field of the uncorrupted frames for the corrupted frames (as both should estimate the ground-truth). Because we agree with your sentiment that this is not a useful definition for robustness, we opted for Lipschitz robustness in RobustSpring. The previously discussed Lipschitz robustness assumes that robust methods should change their predictions only slightly for slight image changes - a subtle but important difference in definitions.
>
> To summarize the robustness discussion for RobustSpring, we purposefully opted for the Lipschitz-based, ground-truth-free view on robustness because (1) it is a widely accepted mathematical definition of model robustness in the literature; (2) it enables the same robustness evaluations regardless of the optical-flow definition that is used (true vs. apparent motion) as it only looks at the strength of input deviation and the resulting model output deviation; and (3) it effectively quantifies if models are able to ignore visual disturbances and maintain their estimate of the overall scene motion.
>
> There is one more question that, though not explicitly asked, reverberates through the review. The question of how a perfectly accurate method would perform in terms of robustness, and what its result means for the utility of the overall benchmark. As mentioned in the review, a perfectly accurate method would score as perfectly robust ($R_{epe}=0$) for certain corruptions (e.g. gaussian blurs and noises) while it would receive imperfect robustness scores ($R_{epe}>>0$) for others (e.g. snow, elastic transform) because it 'correctly' predicts the changed motion field. This behavior is a consequence of disentangling robustness and accuracy: By the above definitions of both quantities, accuracy and robustness _may_ be aligned for certain corruptions, but can also be diametric for others. These behaviors only become possible through their disentanglement. This does not affect the validity of our benchmark for two reasons:
> (1) Independent of the perfectly accurate method's performance, a perfectly robust method will always receive either the same or a lower robustness score. Therefore, the robustness benchmark always highlights particularly robust methods.
> (2) Robustness measures a valuable model quantity, particularly in cases where perfect accuracy and perfect robustness _are not aligned_. In those cases, a robust method is one that still focuses on the important background information and maintains its prediction despite the visual foreground disturbances. One could even argue that if RobustSpring only included corruptions for which perfect accuracy and perfect robustness are aligned, its utility would be reduced considerably, as few new insights over the plain Spring benchmark were added. An instance of those insights is the snow-evaluation experiment in Fig. 5a, where we show that all models propagate detected rain motion into the background and incorrectly estimate the actual scene motion as a result of this weather effect.
>
> We hope that these additional explanations help clarify why RobustSpring's corruption robustness metric is a valid and direct extension of previous robustness measures in the literature, and independent of the optical flow definition. Please let us know if this addresses your concerns. If so, we will use the opportunity to introduce an in-depth discussion to the supplementary material and re-sharpen the metric discussion in the main paper.
>
>
>
> Minor comments
>   *  In the field of optical flow, most currently available unsupervised methods suffer from reproducibility issues (e.g., for SMURF, the results on KITTI/Sintel cannot be reproduced with the author-provided checkpoints). Hence, we opted against including these methods in our evaluation.
>   * We fixed Fig. 2a Elastic tranform in the revised version of our paper.
>   *  While we cannot fully prevent the use of the presented corruptions as data augmentation, we have not provided our code to discourage these augmentations during training. Also, augmenting the whole Spring training set with all 20 corruptions requires a significant resource investment (also see our response to Reviewer npBi), which - though no hard barrier against cheating - might also act as an additional safeguard against it.

---

> > ### Comment · Area_Chair_Hc9j · 2025-08-04
> >
> > Dear Reviewer
> >
> > The Author-Reviewer discussion phase is until 6 Aug. The author has submitted a rebuttal. Please feel free to initialize discussions with the authors.
> >
> > Kind regards
> > AC

---

> > ### Comment · Area_Chair_Hc9j · 2025-08-05
> >
> > Dear Reviewers
> >
> > Thanks for contributing to Neurips. We have received new information from the PCs:
> > “Reviewers must participate in discussions with authors before submitting “Mandatory Acknowledgement”. ” “To facilitate discussions, we extend Author-Reviewer discussions by 48h till Aug 8, 11.59pm AoE. ”
> >
> > As informed, please engage in discussions with the authors.
> >
> > Kind regards
> > AC

---

### Note · Authors · 2025-08-15

We thank all reviewers and the AC for the constructive feedback and positive evaluations.  We are pleased by the consensus on the novelty and practical value of our contributions, and also thank the reviewers for their comments, which further improved our paper (through extended evaluations and improved explanations). Already before the discussion, three reviewers (z5SM, xXnY, npBi) recommended acceptance, with npBi assigning a strong accept and xXnY explicitly "recommending acceptance" after the rebuttal.

In the rebuttal, we clarified key points raised during the discussion:

 * **Robustness metric:** Our Lipschitz-based definition is independent of ground truth, complements accuracy, and aligns with established robustness literature. It enables consistent evaluation even when corruptions alter the motion field.
 * **Depth/disparity estimation:** We quantified estimation errors to show that the estimates are sufficiently accurate for the small subset of corruptions that require them, while preventing ground-truth leakage.
 * **Robustness to occluded regions and motion boundaries:** We added evaluations for occluded regions and clarified motion statistics showing Spring offers larger ranges than Sintel/KITTI.

RobustSpring is the first corruption robustness benchmark for dense matching, introducing temporally/stereo/depth-consistent corruptions and a robustness metric complementary to accuracy. The benchmark is already fully integrated into the existing Spring public benchmark site with data, code, and online evaluation available to the community.

As agreed upon by all reviewers, RobustSpring fills an important gap in current evaluation practice. We are committed to RobustSpring's long-term maintenance and are happy to include the additional evaluation metrics suggested by reviewer xXnY, to make it an actively used resource for advancing robust optical flow, stereo, and scene flow methods.

---

### Decision · Program_Chairs · 2025-09-18

**Decision:**

Reject

**Comment:**

This work proposes a robustness benchmark for dense matching tasks, namely scene flow, optical flow, dense stereo. Specifically the benchmark is focusing on the robustness where the input data deteriorates, for example, in bad weather.
The proposed dataset is an extension to the high resolution Spring dataset where various deterioration were added. To validate robustness without ground-truth, a robust evaluation metric is presented, which is based on Lipschitz continuity to disentangle robustness from accuracy. The authors provide extensive results and analyze their novel benchmark/dataset.
While reviewers recognise the value and quality of the proposed benchmark, three reviewers recommend ‘accept’. Reviewer AZjQ recommends ‘reject’. One reviewer does not give the final recommendation but the initial recommendation is ‘accept’.
The key concern is whether the proposed ground-truth free metric is theoretically sound, specifically whether this may break the definition of optical flow in non-rigid deformations. There are intensive discussions among the authors and reviewers. Unfortunately, a consensus was not reached. Reviewer AZjQ maintained the ‘reject’ recommendation.
The ACs have looked into the case. The ACs find all reviewers agree that the paper is “comprehensive and, on a high level, a very good addition to Spring”. As for the remaining concerns, it does not necessarily overthrow the claimed contributions. Therefore the ACs find the paper can be accepted. However, the ACs recommend the authors to add further analysis about the key concern and state the limitations in the camera ready.

===== FINAL UPDATE FROM DB Track PCs ====

The final decision for this paper has been taken by the program chairs after consultation with the SACs. All Senior Area Chairs have ranked papers according to the feedback from the AC during the review process. We decided to leave the original meta-review to reflect the opinion of the AC in light of the initial discussions with reviewers and SAC.